# ToM-SWE: User Mental Modeling for Software Engineering Agents

**Xuhui Zhou** [1 2]   **Valerie Chen** [1 2]   **Zora Zhiruo Wang** [1]   **Graham Neubig** [1 2]   **Maarten Sap** [1]   **Xingyao Wang** [2]

## Abstract

Recent advances in coding agents have made them capable of planning, editing, running, and testing complex codebases. Despite their growing ability in coding tasks, these systems still struggle to infer and track user intent, especially when instructions are underspecified or context-dependent. To bridge this gap, we introduce ToM-SWE, a dual-agent architecture that pairs a primary software-engineering (SWE) agent with a lightweight theory-of-mind (ToM) partner agent dedicated to modeling the user's mental state. The ToM agent infers user goals, constraints, and preferences from instructions and interaction history, maintains a *persistent memory* of the user, and provide user-related suggestions to the SWE agent. In two software engineering benchmarks (ambiguous SWE-bench and stateful SWE-bench), ToM-SWE improves task success rates and user satisfaction. Notably, on stateful SWE benchmark, a newly introduced evaluation that provides agents with a user simulator along with previous interaction histories, ToM-SWE achieves a substantially higher task success rate of 59.7% compared to 18.1% for OpenHands, a state-of-the-art SWE agent. Furthermore, in a three-week study with professional developers using ToM-SWE in their daily work, participants found it useful 86% of the time, underscoring the value of stateful user modeling for practical coding agents.

## 1. Introduction

Recent advances in large language models (LLMs) have enabled coding agents to perform complex software engineering tasks, from code generation (Jiang et al., 2026) and

debugging (Tian et al., 2024) to system design (Kovacic et al., 2025) and optimization (Gao et al., 2024). However, despite their impressive technical capabilities, coding agents often struggle with a fundamental aspect of software development: *effective communication* and *collaboration* with human developers.

The core limitation is that current systems lack explicit mechanisms for modeling and predicting human intent in long-horizon, multi-turn interactions (Kim et al., 2023; Zhou et al., 2024). Unlike human developers who naturally build mental models of their collaborators' goals, preferences, and constraints through various tasks (Tomasello, 2009), coding agents lack the mechanism to infer and acquire the underlying user intentions from the surface-level instructions. Furthermore, current coding agents typically operate in a stateless manner, treating each session as independent rather than maintaining persistent context about the user's evolving goals and conversation history. This paradigm often leads to wasted effort and misunderstandings, and in high-stakes settings can result in erroneous, or even unsafe outcomes.

To bridge the gap between current coding agents and the challenges of inferring user intent in long-horizon interactions, we introduce ToM-SWE, a conceptual framework that integrates *theory-of-mind* (ToM) reasoning into software engineering agents. Here, ToM refers specifically to the ability to model a user's mental state, including goals, preferences, and intentions, based on user instructions and interaction history. As shown in Figure 1, ToM-SWE operationalizes this idea through a dual-agent architecture: a primary software engineering (SWE) agent remains focused on coding tasks, while a dedicated ToM agent models the user's mental state and supports the SWE agent when needed. This separation is crucial for two reasons: it preserves the SWE agent's coding performance, and it enables specialized, persistent user modeling that developers can flexibly invoke and customize for efficiency and privacy. The ToM agent itself functions in two complementary modes to keep track of the user's preferences, emotions, and etc. During *active coding sessions* (in-session ToM), it infers the user's underlying mental state (e.g., the "true" intent behind potentially ambiguous instructions). After each session, it works to create mental models of the user (after-session ToM), consolidating interaction history to refine its beliefs about the user's mental state in a

[1]Language Technology Institute, Carnegie Mellon University [2]All Hands AI. Correspondence to: Xuhui Zhou <xuhuiz@cs.cmu.edu>.

*Proceedings of the 43rd International Conference on Machine Learning*, Seoul, South Korea. PMLR 306, 2026. Copyright 2026 by the author(s).

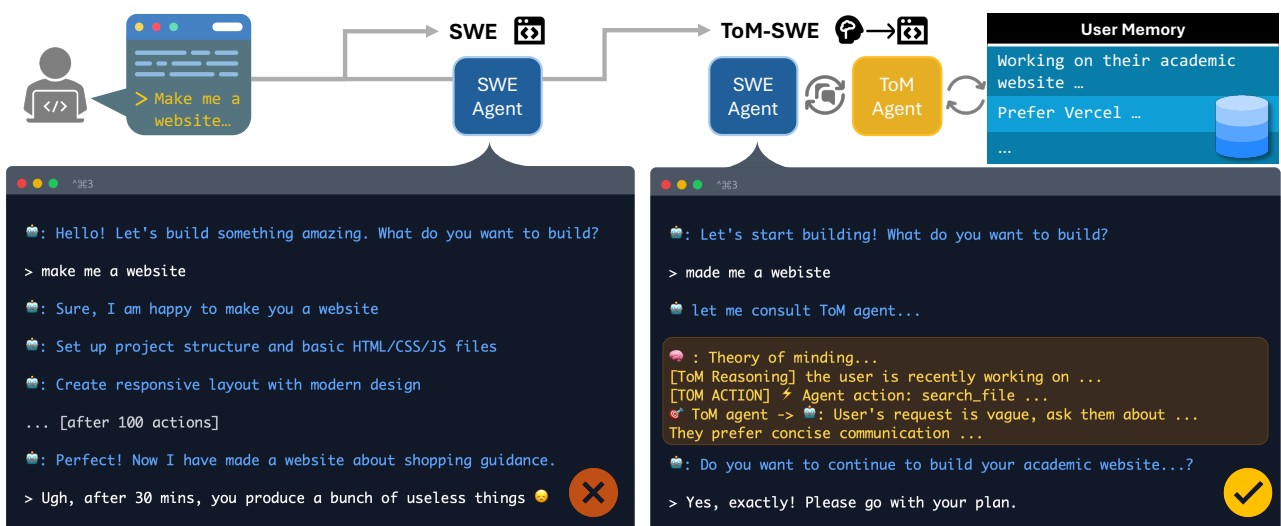

Figure 1. An example of how the ToM-SWE framework can help the SWE agent to model the user's mental state and provide more useful suggestions based on the previous interaction history. When facing **the same** starting instruction ("Make me a website..."), the SWE agent simply generates code yet fails to meet the users' requirement for the task. However, with the ToM agent, the SWE agent can first consult the ToM agent that maintains a persistent model of the user mental state across multiple sessions, and then act more aligned with the user's preferences and constraints.

hierarchical way.

To enable evaluation of coding agents in real-world settings, where agents must adapt to different users over long-horizon, multi-session interactions to solve software engineering tasks. We therefore introduce the *Stateful SWE benchmark*, the first benchmark that allows agents to leverage realistic conversation histories across multiple coding sessions to track user mental state. In this setting, agents interact with an LLM-powered user simulator and receive synthesized interaction histories to guide their reasoning. To stay consistent with prior evaluation, we borrow the original issues from SWE-bench (Jimenez et al., 2024), reframing them as ambiguous starting instructions paired with different user profiles and interaction histories. The agent needs to understand user intent, preferences and constraints either through interacting with the LLM-powered user simulator or inferring from the past interaction histories. Unlike existing benchmarks such as SWE-bench, which primarily assess technical problem-solving, Stateful SWE evaluates an agent's ability to sustain meaningful interactions over time.

We evaluate our ToM-SWE framework on both our newly introduced stateful SWE benchmark as well as the stateless Ambiguous SWE-bench (Vijayvargiya et al., 2026). We build our agent (`ToMCodeAct`) using the OpenHands platform (Wang et al., 2025), an open-source framework for developing SWE agents. We show that `ToMCodeAct` outperforms the OpenHands SOTA `CodeAct` agent (Wang et al., 2024) on both benchmarks. On the ambiguous SWE-bench, `ToMCodeAct` agent achieves 63.4% issue resolved rate compared to CodeAct agent's 51.9% (+11.5% improvement). On the stateful SWE-bench, `ToMCodeAct` agent achieves 57.4% (+43.9% task resolved rate compared to CodeAct agent's 13.5%. Furthermore, `ToMCodeAct` agent achieves substantially better user satisfaction scores of 3.62 compared to CodeAct agent's 2.57 (+41% improvement; automatically measured through user simulators that evaluate preference alignment, communication, etc.). The results highlight the importance of modeling user mental state in real-world software development, especially when user instructions are high-level and underspecified and preferences and constraints are not stated explicitly.

Finally, we conduct a three-week *in-the-wild* observational human study with 17 professional developers using a ToM-enhanced OpenHands CLI (Wang et al., 2025) on their everyday, self-chosen coding tasks (rather than predefined tasks as in prior work (Wu et al., 2025; Qian et al., 2025a)). Our primary goal is to evaluate the interaction-level utility of ToM suggestions in professional workflows: each time the system produces a ToM suggestion, developers mark it as `Accept`, `Almost right, let me modify it`, or `Reject`. Across 209 sessions (174 ToM suggestions), developers accept or partially accept ToM suggestions 86% of the time. Besides the high acceptance rate, we learn from the developers' feedback that ToM agent can often provide useful and valuable suggestions that make users workflow more efficient (e.g., *"Please add pytest for the new function"* or *"keep your code edit minimal"*). Furthermore, ToM agent can even provide novel and preference aligned suggestions (e.g., *"refactor the code following Linux development philosophy"*). Our work sheds light on building more proactive

and personalized software engineering agents that can adapt to different users and tasks.

## 2. Related Work

**Software Engineering Agents** Large language models enable competitive AI agents for software engineering. Systems like SWE-agent (Yang et al., 2024), CodeAct (Wang et al., 2024), demonstrate impressive automation capabilities through agent-computer interfaces and executable code actions (Jimenez et al., 2024). The standard benchmark for software engineering agents is SWE-bench (Jimenez et al., 2024), which evaluates agents' ability to handle well-specified software engineering tasks, and use unit tests to decide the task success. However, these systems only interact with environments and overlook human developer interaction. Recently, ClarifyGPT (Mu et al., 2023) addresses requirement ambiguity through two-step consistency checks but focuses only on simple function generation. Vijayvargiya et al. (2026) introduce Ambiguous SWE-bench, showing that interaction helps agents handle underspecified software engineering tasks. However, as the benchmark operationalizes ambiguity primarily by underspecifying the current instruction, it does not capture the long-term memory and preference-tracking challenges that arise in multi-session settings.

**Theory of Mind and Personalization in AI Systems** Theory of mind (ToM) is crucial for AI systems engaging with humans.[1] Li et al. (2023) show that LLM-based agents with ToM capabilities better coordinate in multi-agent collaboration, essential for complex software development (Qian et al., 2024). While popular benchmarks like SOTOPIA (Zhou et al., 2024) evaluate social intelligence and FANToM (Kim et al., 2023) stress-test machine ToM, the ToM abilities of software engineering agents remain underexplored despite requiring close human-agent collaboration. Similarly, many personalization techniques have been proposed, including personalized reinforcement learning (Li et al., 2024), parameter-efficient conversation personalization (Magister* et al., 2025), user embeddings (Ning et al., 2024). However, how to effectively build personalized SWE agents is still an open question.

**Agent Memory Systems** Effective memory management is critical for long-term AI agent interactions. RAG systems suffer from context pollution and fail to capture nuanced information (Letta, 2024). Recent advances include MemGPT (Packer et al., 2024) with dual-tier memory hierarchies, Mem0 (Chhikara et al., 2025) achieving faster

context retrieval through dynamic consolidation, and MemoRAG (Qian et al., 2025b) addressing context pollution via draft generation. However, existing systems focus on general conversation contexts, leaving a gap in memory architectures for software engineering where agents must maintain complex mental models of user preferences, coding styles, and evolving requirements across sessions.

## 3. ToM-SWE: Pairing SWE Agent with ToM Agent

Consider existing setups of software engineering agents such as SWE-agent (Yang et al., 2024) and OpenHands CodeAct (Wang et al., 2024; 2025). At time step $t$ in coding session $i$, an agent receives an observation $o_t \in O$ either from the environment (e.g., terminal output, file contents, test results) or the user (e.g., user instructions, feedback). Then the agent takes an action $a_t \in A$ (e.g., code edits, shell commands) following some policy $\pi(a_t|c_t^i)$, where $c_t^i = (o_1, a_1, \ldots, o_{t-1}, a_{t-1}, o_t)$ is the context available to the agent in the coding session $i$ until time step $t$.

The mapping $c_t^i \mapsto a_t$ becomes challenging when accurate execution of the tasks requires understanding implicit user preferences as such information often exist in past sessions $\{c^j\}_{j=1}^{i-1}$, instead of the current context $c_t^i$. For example, when a user says *"implement a web scraper"*, the agent needs to infer library preferences (`requests` vs `httpx`), which could only be available from previous interactions. To bridge this gap, we introduce a theory-of-mind (ToM) agent that explicitly models the user's mental state (Figure 2). In the following, we explain (1) how the SWE agent queries the ToM agents for relevant information and (2) how the ToM agent models the user's mental state.

**SWE Agent Interaction with ToM Agent** We enable the SWE agent to interact with the ToM agent by introducing two additional tools into its available toolset: (1) `consult_tom` (*in-session*): the SWE agent sends a query $q$ and the current session context $c_t^i$ to the ToM agent (action $a_t$), the ToM agent outputs relevant user mental state information $m_{user}$ by reasoning over the interaction history $\{c^j\}_{j=1}^{i-1} \cup \{c_t^i\}$. This user modeling information is then incorporated into the SWE agent's context as $c_t \| [a_t, m_{user}]$, helping the agent to make decisions that aligns with the user's implicit preferences and constraints. (2) `update_memory` (*after-session*): After the SWE agent finishes the coding session, it uses this tool to inform the ToM agent to process the current session and update the hierarchical memory system.

**ToM Agent Design** The ToM agent models user mental states through a three-tier hierarchical memory system implemented as external database: (tier 1) raw session storage,

---

[1]We use "ToM" in a deliberately practical sense: our ToM agent operationalizes user mental modeling and is closely related to, rather than distinct from, memory- and preference-modeling approaches.

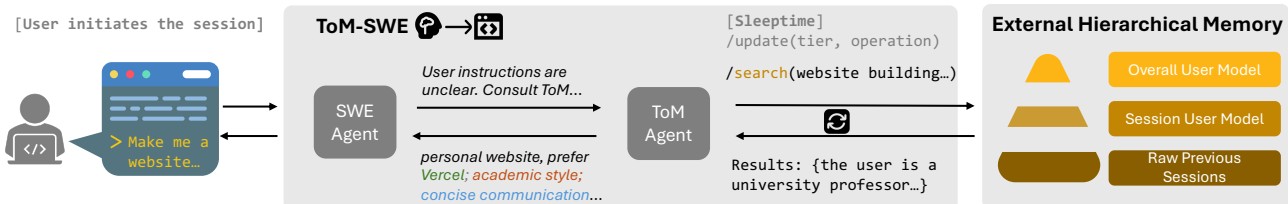

*Figure 2.* Overview of the ToM-SWE framework: the SWE agent handles code generation and execution, while the ToM agent focuses on user modeling and intent inference. The SWE agent consults the ToM agent to predict the user's mental state before suggesting technical actions. Meanwhile, the ToM agent maintains an external hierarchical memory system to persist the user's state and update user models after each session (with `update_memory` action).

stores complete previous session histories. (tier 2) session-based user model, maintains per-session analysis including session intent, interaction patterns, and coding preferences. (tier 3) overall user model, aggregates cross-session patterns into preference clusters, interaction style summary, and coding style summary.

During the *in-session* phase, once the SWE agent sends the query and current session history $(c_t^i)$ to the ToM agent, the ToM agent with overall the user model loaded in the context window could decide to use the `search_file` action to retrieve the relevant context or use the `read_file` action for a specific file from the (tier 1) and (tier 2) of the memory system. The retrieved/read content will be added to the context window of the ToM agent. The ToM agent can perform multiple actions to obtain the relevant information before providing suggestions to the SWE agent with `give_suggestions` action.

During the *after-session* phase, the ToM agent processes new session data through a structured workflow. Raw sessions are first added to (tier 1) automatically. The ToM agent then uses `analyze_session` with raw session data as the input, extracting user intent, emotional states, and message-level preferences to create structured session-based user models (tier 2). If there's no overall user model, ToM agent will use `initialize_user_model` to aggregate these session-based user models to update the overall user model (tier 3). If there's already an overall user model, ToM agent will use `update` action to update the overall user model. (See Appendix A.4 for more action space details.)

This dual-agent design offers two advantages over having the SWE agent handle user modeling directly: (1) *reduced context distraction*: the SWE agent maintains focus on technical tasks without being overwhelmed by extensive user history, (2) *specialized optimization*: each agent can be optimized for its specific domain (coding vs. user modeling)

## 4. Stateful SWE Benchmark

Most of previous SWE benchmarks solely focus on task completion (Zhao et al., 2024; Zan et al., 2025; Chowdhury et al., 2024). Here, we introduce a new benchmark that extends SWE-bench (Jimenez et al., 2024) to test agents' ability to interact, model and adapt to users while solving tasks. For each instance in our benchmark, agents not only have access to the coding environment, but can also interact with the simulated user, and check the previous session history with the user.

**Profile Collection:** As shown in Figure 3, we begin by collecting 453 real, consented sessions between human software developers and coding agents through the OpenHands platform (Wang et al., 2025). From the collected sessions, we derive 15 developer profiles capturing distinct interaction styles (verbosity, question timing, response style) and coding preferences (testing practices, documentation habits, architectural choices). Each profile represents a unique combination of interaction patterns paired with coding preference clusters derived from 75 recurring practices observed in the sessions (see Appendix A.6 for detailed breakdown). For each profile, we randomly sample 20 sessions from the same developer to form the profile's history. To maintain realism, these sessions are processed with GPT-5 to align user messages with their corresponding profile characteristics (e.g., rephrasing a verbose user message into a concise one if the profile prefers concise exchanges).

**Instance Creation and Evaluation:** We pair created 15 developer profiles with 500 instances from the verified SWE-bench issues (Chowdhury et al., 2024) and run a user simulator powered by LLM that enables realistic human-agent interaction evaluation, inspired by Ambiguous SWE-bench Vijayvargiya et al. (2026). Differing from Ambiguous SWE-bench, the user simulator in Stateful SWE-bench is conditioning on the unique user profile with interactional and coding preferences. Therefore, simulators with different profiles behave differently, posing challenges for the agent to interact with different users. For example, a user with low verbosity preference will only answer one question and

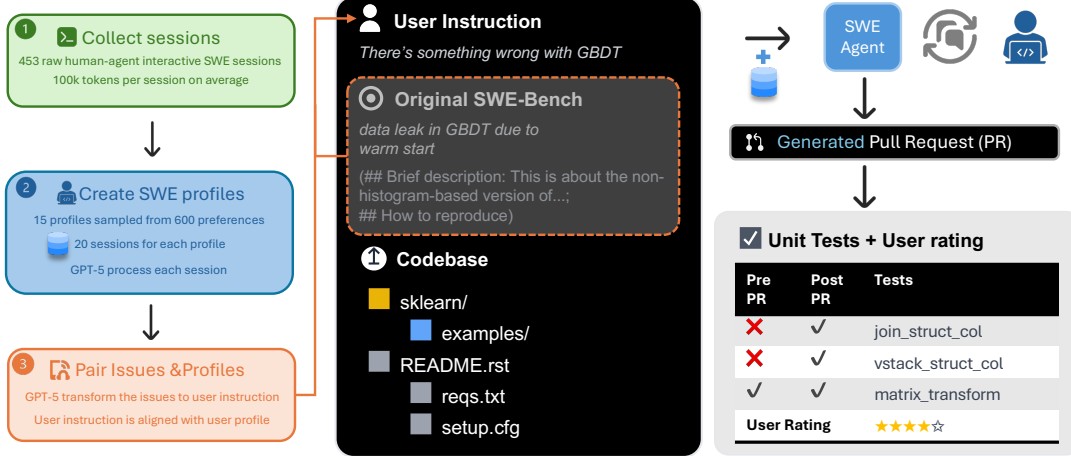

*Figure 3.* Overview of the stateful SWE benchmark. We first **collect profiles** following the three steps outlined above. We then **create instances** by pairing the user simulator of different profiles with SWE-bench issues. Original SWE-bench issue descriptions are converted into vague user instructions (similar to Ambiguous SWE-bench (Vijayvargiya et al., 2026)), but our user simulator conditions on personalized profiles with distinct interaction styles and coding preferences, whereas Ambiguous SWE-bench uses fixed user simulator prompts. The agent solves the task under the same environment and tests of the original instance with access to the previous interaction histories with the user and could interact with the simulated user for extra information.

could express dissatisfaction or even refuse to answer if the agent asks too many questions in a single turn. Meanwhile, agents have access to previous conversation histories with the same user profile, requiring them to derive user preferences from past interactions to communicate successfully. Besides the standard SWE-bench instance evaluation, we could also apply the user simulator to evaluate the agent's ability to interact with the user. We give the full session data to the corresponding profile-conditioned user simulator and ask the user simulator to rate the agent from 1 to 5 and obtain the *user simulator satisfaction* scores (See Appendix A.8.2 for details).

## 5. Offline Benchmark Experiments

### 5.1. Experiment Setup

We implement all experiments using the OpenHands platform (Wang et al., 2025), an open-source framework for developing and evaluating AI software development agents. OpenHands provides sandboxed environments for safe code execution and standardized interfaces for agent-environment interaction while maintaining isolation and reproducibility. Our experiments build upon the `CodeAct` agent architecture (Wang et al., 2024), which uses executable code as a unified action space for agent interactions `CodeAct` consolidates traditional agent actions (e.g., file editing, command execution, web browsing) into executable code snippets that are dynamically interpreted within the environment.

**Baselines and Setup**    Our experiments involve three agent variants: (1) `CodeAct` agent: The baseline implementation

following Wang et al. (2024), which operates without explicit user modeling capabilities. (2) `RAGCodeAct` agent: An enhanced version that encourages the coding agent to proactively retrieve relevant information from previous interaction history, serving as the single agent paradigm to manage both the coding task and user modeling task simultaneously. (3) `TomCodeAct` agent: Our proposed approach that pair coding agent with a theory-of-mind agent, maintaining explicit user mental models and adapting behavior accordingly. We evaluate all agents using three state-of-the-art language models: Claude Sonnet 4, Claude 3.7 Sonnet and Qwen3-480B (Qwen3). All agents interact with the same simulated users and are prompted to ask for clarifications when uncertain. We use Claude Sonnet 4 for the theory-of-mind agent across all experiments and use BM25 to retrive relevant information for both the `RAGCodeAct` agent and the `TomCodeAct` agent (see Appendix A.8 for complete model specifications and hyperparameters).

**Evaluation Benchmarks**    As described in Section 4, we evaluate our approach on two complementary benchmarks: the Ambiguous SWE benchmark and our Stateful SWE benchmark. Both benchmarks use GPT-5 powered simulated users and evaluate agents over 500 instances with a maximum of 100 interaction turns per task. Note that the user simulator has access to both the complete issue description and the hints for the issue from the original SWE-bench issues. Together, these benchmarks evaluate complementary scenarios: ambiguous formal specifications requiring clarification versus informal user instructions with available context. We report task resolved rates (i.e., the percentage of tasks passed the unit tests provided in the SWE-bench

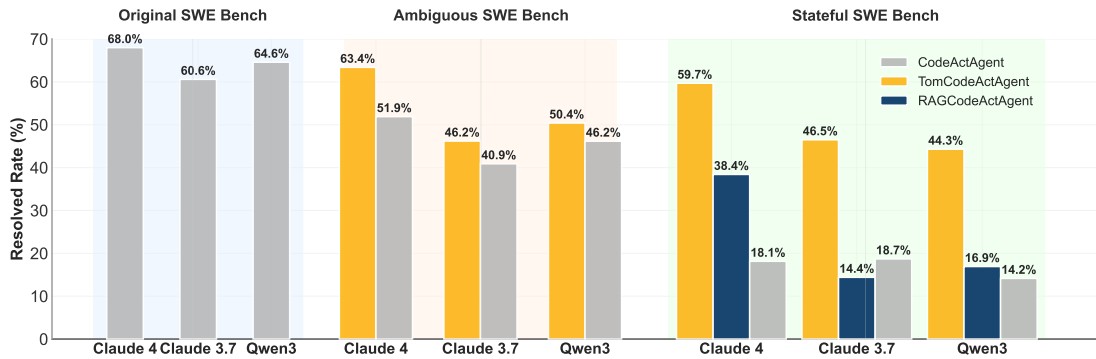

*Figure 4.* Performance comparison based on task resolved rates (measured by passed unit tests provided in the SWE-bench issues). `TomCodeAct` agent consistently outperforms `CodeAct` agent across both benchmarks and model variants, with the largest performance gap observed in the Stateful SWE benchmark using Claude Sonnet 4.

issues) for both benchmarks and additionally report user simulator satisfaction scores for the Stateful SWE benchmark (i.e., the satisfaction scores generated by the LLM user simulator) [2].

### 5.2. Benchmark Results

We present evaluation results demonstrating ToM-SWE's effectiveness for both benchmarks and human study. Our findings show consistent improvements in both task resolution rates and user satisfaction when agents incorporate theory of mind capabilities for user modeling.

Figure 4 shows the resolved rates across all model-agent combinations. For example, `TomCodeAct` agent maintains its lead with 63.4% resolution rate using Claude Sonnet 4 versus `CodeAct` agent's 51.9% on the Ambiguous SWE benchmark. And `TomCodeAct` agent achieves 59.7% resolution rate with Claude Sonnet 4 compared to `CodeAct` agent's 18.1%, representing a 41.6 percentage point improvement on the Stateful SWE benchmark. Table 1 shows

*Table 1.* User Simulator Satisfaction Scores on Stateful SWE Benchmark

| Agent | Claude 3.7 | Claude 4 | Qwen3 |
|---|---|---|---|
| CodeAct | $2.26_{\pm 0.08}$ | $2.57_{\pm 0.08}$ | $2.48_{\pm 0.08}$ |
| +RAG | $2.32_{\pm 0.08}$ | $3.09_{\pm 0.09}$ | $2.54_{\pm 0.11}$ |
| +ToM | $\mathbf{3.29}_{\pm 0.08}$ | $\mathbf{3.62}_{\pm 0.07}$ | $\mathbf{3.24}_{\pm 0.09}$ |

user satisfaction scores for the Stateful SWE benchmark. `TomCodeAct` agent achieves the highest satisfaction ratings across all models [3].

---

[2]Please refer to the Appendix A.10 for more details on the scoring process of user simulators.

[3]Note that we tested a single-agent baseline that manages both coding tasks and ToM-related contexts. In preliminary experiments (N=100) with Claude Sonnet 4, this setup achieved only a 27.17% resolution rate on the stateful SWE bench. We attribute this un-

Besides the success of the `TomCodeAct` agent, we additionally find that Theory-of-mind reasoning over past user-agent interactions is essential for understanding current user intent while simply retrieving raw session data provides limited help. We observe that while `RAGCodeAct` agent also has access to the previous raw session data, it still underperforms the `ToMCodeAct` agent across all models. This is especially true when the base model for SWE agent is not powerful enough to handle both the coding task and the user modeling task simultaneously. With Claude 3.7 Sonnet, `RAGCodeAct` agent even hurts the performance of the SWE agent (task resolved rate drops from 18.7% to 14.4%).

**Relationship between Task Resolution and User Satisfaction** We conduct regression and correlation analyses to understand the relationship between task resolution rate and user satisfaction. We find a strong positive association: for the Claude 4 `ToMCodeAct` agent, the Pearson correlation coefficient between resolution and satisfaction is $r = 0.74$ ($p < 0.001$), and task resolution explains 55% of the variance in satisfaction scores ($R^2 = 0.55$). However, we also observe cases where the agent fails to resolve the task but user satisfaction is high (F+H)[4]. Specifically, we bucket satisfaction scores into *High* (3.5–5), *Medium* (2–3.5), and *Low* (1–2), and focus on this key disagreement case (F+H). Table 2 reports F+H rates across agent–model configurations. We find that the `ToMCodeAct` agent consistently achieves

---

derperformance to the difficulty of managing high-complexity contexts within a single agent and did not extend the baseline further due to budget constraints.

[4]We also observe cases where tasks are resolved but satisfaction is only medium (S+M) due to preference violations or communication style (e.g., "Ignored the preferred typing style... Did not provide a descriptive commit message" and "You didn't provide a brief summary..."). However, we find these cases happen less frequently (3% of the time), and there's no statistically significant difference of the S+M rates among different agent-model configurations.

*Table 2.* Task resolution and user satisfaction disagreement rates (in %). F+H: failed to resolve the task but user satisfaction is high (higher is better).

| Model | CodeAct F+H↑ | ToMCodeAct F+H↑ | RAGCodeAct F+H↑ |
|---|---|---|---|
| Claude 3.7 | 3.7 | **19.3** | 5.0 |
| Claude 4 | 8.8 | **21.5** | 14.0 |
| Qwen3 | 5.9 | **15.4** | 8.0 |

the highest F+H rates, suggesting a "good-effort bonus" where users reward strong process quality and preference alignment even without a final fix. Qualitatively, in F+H cases the simulator often credits meaningful progress and good communication (e.g., "precise fix with minimal back-and-forth, aligned well with my preferences" and "Asked for all key details up front (version, minimal repro, affected APIs), matching the preferred workflow"). To understand F+H cases in more detail, we manually examine 44 instances where the Claude 4 `ToMCodeAct` agent received high satisfaction despite failing the task; users were rewarding process quality rather than being misled about outcomes. Specifically, users valued (a) systematic, well-targeted investigation with clear technical reasoning (61% of F+H cases) and (b) clear communication and visible progress toward a solution (39% of F+H cases).

### 5.3. Cost Efficiency of ToM Agent

For each ToM base model (GPT-5 *nano*, GPT-5 *mini*, GPT-5, Claude 3.7, Claude 4), we run the `TomCodeAct` agent on 100 sampled Stateful SWE-bench instances. We then report the resolved rate (%) alongside the average session cost (USD), which includes the full end-to-end cost of using the ToM agent during problem solving. As shown in Figure 5, even very efficient ToM base models substantially improve performance while adding only a small fraction of the overall session cost. For instance, GPT-5 *nano* reaches 38.0% at only $0.02 per session. The top axis reports ToM overhead as a fraction of the average SWE cost per session ($1.08); for example, using Claude 4 as the ToM base adds $0.17 per session, i.e., about 16% overhead.

### 6. In-the-wild Human Study

To assess the real-world utility of `TomCodeActAgent`, we conducted a three-week observational field study with 17 software developers. Unlike a controlled laboratory experiment, this "in-the-wild" deployment allowed participants to interact with the agent within their familiar CLI environment while working on their own unscripted, day-to-day coding tasks.

The primary objective was to evaluate the interaction-level utility of the ToM agent's suggestions in a professional context. Given the diverse and private nature of the participants'

work, we prioritized an observational design over a comparative control-group study to maximize ecological validity. Consequently, we focus our analysis on user-acceptance metrics, specifically, whether developers accepted, modified, or rejected the ToM agent's suggestions. Here we don't broad causal claims about productivity improvements.

In total, we recorded 209 coding sessions. Within these sessions, the ToM agent proactively provided 174 suggestions based on the inferred user state. Due to strict privacy constraints, we did not collect codebase content or specific dialogue, focusing exclusively on the developers' behavioral responses to the agent's interventions.

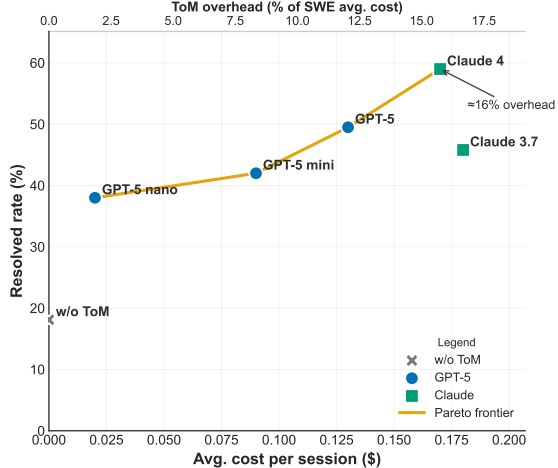

*Figure 5.* Resolved rate vs avg. ToM cost per session in stateful SWE benchmark. The orange line highlights the Pareto frontier: configurations that are not dominated by another option with both lower cost and higher resolved rate. The top axis shows ToM overhead as a percentage of the typical SWE session cost ($1.08) per session.

**Study Design and Success Metrics**   Our evaluation focuses on practical utility: we measure success as the rate at which developers accept (fully or partially) ToM agent suggestions during real coding work. Participants install the ToM-enhanced coding agent CLI and use it for their daily coding tasks over three weeks [5]. During a session, developers work in the CLI workflow while the SWE-agent attempts to solve the task; the SWE-agent decides whether to consult ToM, and if so, the ToM agent summarizes the developer's likely intent and preferences and proposes concrete suggestions (e.g., user preferred tools, user preferences, etc.) according to previous interaction history. After the session ends, the ToM agent automatically reflects on the session

---

[5]We adopt the open-source OpenHands CLI https://github.com/OpenHands/OpenHands as the coding environment for the human study.

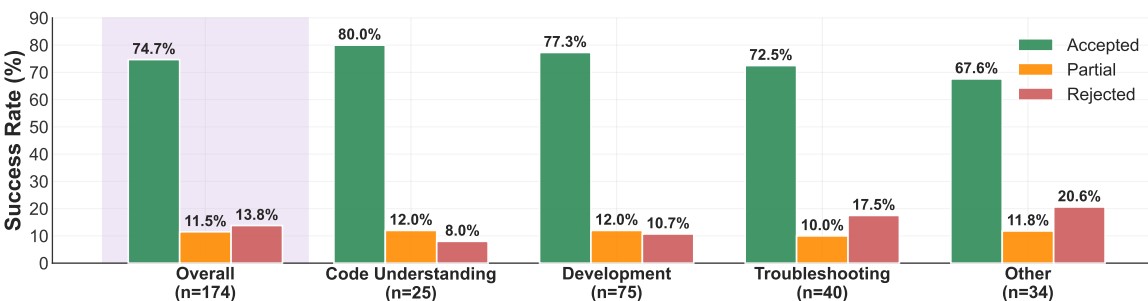

*Figure 6.* ToM consultation analysis across 174 human study interactions. The overall success rate of 86.2% varies by query category, with Code Understanding achieving 92% success while Other queries succeed only 79.4% of the time.

and updates the user profile for future sessions [6].

When the ToM agent provides suggestions, participants choose from three options: (1) `Accept`, use ToM's suggestions directly, (2) `Almost right, let me modify it`, combine ToM's suggestion with participant modifications, or (3) `Reject`, proceed with the original instruction. This annotation process captures real-time user preferences and validates the ToM agent's understanding of user intent. The participants could also provide feedback on the ToM agent's behavior throughout the study.

As shown in Figure 6, developers find the ToM agent's suggestions useful, with an overall success rate of 86.2% (combining 74.1% full acceptance and 12.1% partial acceptance). Success varies by query category: Code Understanding achieves the highest acceptance (80.0% + 12.0% = 92.0%), followed by Development (75.2%), Troubleshooting (82.5%), and Other tasks (79.4%).

From the user feedback, we often find users would be happy to use the ToM agent's suggestions to help them guide the SWE agent: *"I find these suggestions helpful, ToM helps me explicitly write out rules I already have in my previous conversations."*, *"I feel ToM agent creates a accurate user profile for me."*, and *"It's more efficient now with the help of ToM agent."*.

To better understand where ToM agent succeeds and fails, we randomly sample 50 suggestions and analyze them in detail and have the following observations:

(1) **Context Specificity Spectrum.** ToM agents excel with moderately underspecified queries that have sufficient technical context, such as *"User wants to refactor ConversationStats to be a Pydantic class and integrate it with ConversationState."* Here, the ToM agent leverages previous conversations to provide useful suggestions. However, when queries become extremely vague (e.g., using `/tom_give_suggestions` without specific context),

---

[6]Figure 7 shows a concrete example where ToM learns persistent preferences across sessions and proactively prevents coding-standard violations.

success rates drop significantly. This suggests ToM agents can handle a spectrum of ambiguity but struggle with highly underspecified scenarios

(2) **Confidence Correlates with Acceptance.** Successful consultations typically exhibit 90-95% confidence levels, while failures often show lower confidence (e.g., 70%). When ToM agents are uncertain about user intent, they provide generic suggestions that fail to match user expectations, particularly in Troubleshooting scenarios (82.5% success rate).

These findings show that effective AI user modeling in software engineering requires not just reasoning capabilities, but also actively engage with the user with proper confidence levels. And the challenge of how to leverage user efforts in this collaborative development workflow remains an open question for future work.

## 7. Discussion & Conclusion

Our results suggest that effective user modeling for software engineering agents benefits from (i) *explicit* representations of user intent and preferences and (ii) *dedicated* reasoning capacity to maintain and apply these representations during long-horizon problem solving. Across both offline benchmarks and the in-the-wild human study, ToM-SWE improves not only task success but also the interaction experience, indicating that preference alignment and process quality matter alongside unit-test outcomes.

An important takeaway is that simply retrieving prior interactions is often insufficient: the agent must also abstract raw user interactions into concrete constraints (e.g., tool choices, style conventions, and communication norms) while still maintaining strong code reasoning. This motivates our multi-agent design, which separates user modeling (ToM) from task execution (SWE) to reduce interference and context overload.

In this work, we rely on an LLM-powered user simulator to assess interaction quality, which is cost-effective but may

introduce systematic biases (e.g., over-knowledgeability and excessive compliance) (Lin & Tomlin, 2025); we mitigate this via iterative calibration and manual verification (Appendix A.8.2). We also identify that personalized user modeling raises privacy and consent considerations (e.g., on-device ToM, user-controlled memory), and generalization beyond software engineering to other collaborative domains remains an open direction.

**Conclusion**    We presented ToM-SWE, a framework that augments software engineering agents with theory-of-mind capabilities for modeling and adapting to individual users. Across two interactive benchmarks and a real-world study with professional developers, ToM-SWE improves both task resolution and interaction quality, showing that strong human–AI collaboration requires agents that can infer, maintain, and act on user-specific mental models over time.

## Impact Statement

This work studies a dual-agent framework (ToM-SWE) that models user preferences and mental state to improve software engineering assistance. If deployed responsibly, such user-aware agents could increase developer productivity and reduce friction in interactive debugging and maintenance. However, user modeling also raises risks that must be handled carefully.

**Human Subjects, Privacy, and Data Protection**    We collected 453 consented developer sessions from the Open-Hands platform with explicit user consent for research purposes. All user data was anonymized by removing identifying information (e.g., usernames, email addresses, repository names, and personal file paths). Session data was processed to remove sensitive information such as API keys, passwords, or proprietary code snippets. Our design additionally supports privacy-preserving deployment patterns (e.g., keeping ToM-related memory on-device while the SWE agent runs in the cloud) and can incorporate differential privacy mechanisms when appropriate.

**Potential for Misuse and Mitigations**    Capabilities for modeling users could be misused for manipulation or unauthorized surveillance. We emphasize that our system is designed specifically to improve software development assistance, not for psychological profiling. We recommend deployments include (i) explicit user consent, (ii) transparent disclosure of what is stored and how it is used, (iii) minimal data retention with user-controlled deletion, and (iv) access controls around any persistent memory.

**Bias and Fairness**    Our 15 developer profiles derived from OpenHands sessions may not represent the full diversity of software developers globally. They are predominantly based on English-speaking developers using particular languages and tools, which may reduce performance for underrepresented populations. Future work should broaden participant diversity and incorporate explicit fairness evaluations for user modeling.

**Reproducibility**    We provide comprehensive implementation details in the Appendix, including (i) prompt templates (Appendix A.7), (ii) memory system architecture and JSON schemas (Appendix A.9), (iii) algorithmic specifications (Algorithm 1), and (iv) experimental configurations, hyperparameters, and evaluation procedures (Appendix A.8). Due to privacy constraints, the original 453 developer sessions cannot be released; instead we provide anonymized developer profiles (Table 3) and synthetic examples illustrating the data formats (Appendix A.9).

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

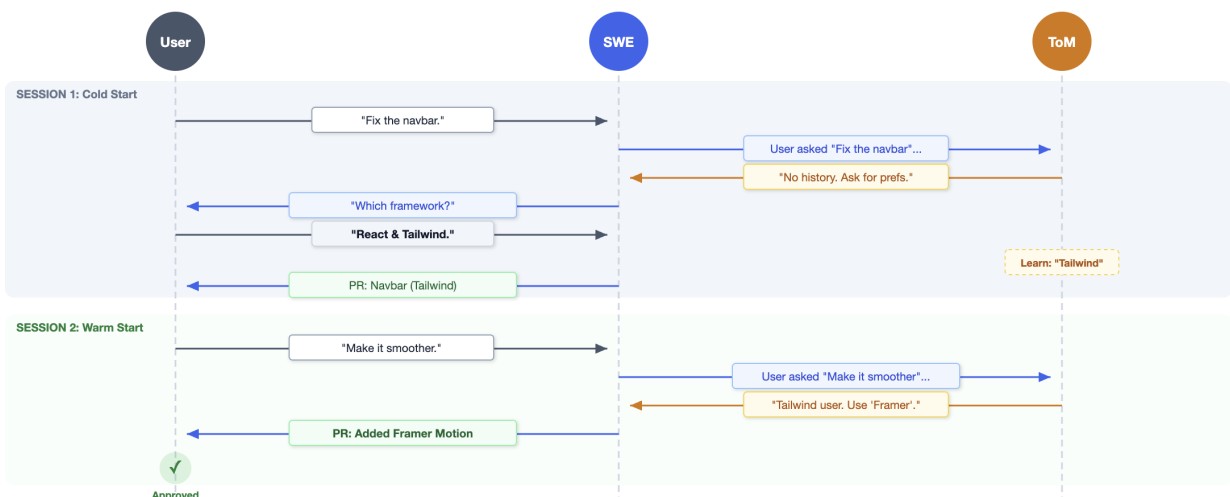

*Figure 7.* Interaction timeline showing how the ToM agent enables preference-aware development in real-world usage across multiple sessions. The cold start is the first interaction with the ToM agent, and the warm start is the subsequent interactions with the ToM agent after the ToM agent learns more about the user. The green checkmark indicates successful task completion aligned with user preferences. Content is shortened for illustration purposes.

## A. Implementation Details

This appendix provides comprehensive implementation details to enable full reproduction of the ToM-SWE method. Section A.1 addresses LLM usage in our research. Section A.2 presents the algorithmic specifications and action details for the ToM agent. Section A.6 contains the five core prompt templates that implement our prompting methodology. Sections A.7-A.9 detail the experimental configuration, data formats, and system architecture respectively.

### A.1. LLM Use in Research

We only use large language models for language-related assistance such as polishing clarity, grammar, and readability. For example, they were occasionally used to rephrase sentences for smoother flow, check for consistency in terminology, or simplify overly complex phrasing.

## A.2. Additional Human Study Example

## A.3. ToM-SWE Algorithm

---

**Algorithm 1** ToM-SWE Agent (in-session and after-session Operations)

---

 1: **function** SWE-Agent($instruct$)
 2:     $h_t \leftarrow []$
 3:     **loop** {}
 4:         $action \leftarrow$ generate_act($instruct, h_t$)
 5:         **if** $action$ is consult_tom **then** {}
 6:             $sug \leftarrow$ ToM-Agent($instruct, h_t$) {in-session consultation}
 7:             $action \leftarrow$ adapt($action, sug$)
 8:         **end if**
 9:         $obs \leftarrow$ execute($action$)
10:         $h_t \leftarrow h_t \cup \{action, obs\}$
11:         **if** $action$ is finish **then** {}
12:             **break**
13:         **end if**
14:     **end loop**
15:     update_memory($h_t$) {after-session memory update}
16: **end function**

---

## A.4. ToM Agent Action Specifications

The ToM agent implements a structured action space through the `ActionExecutor` class, which provides eight distinct actions organized into three categories: file operations, memory system updates, and response generation. Each action is type-safe using Pydantic models and supports both in-session consultation and after-session memory processing workflows. We limit the number of actions to 3 before giving the suggestions to the SWE agent by default for efficiency.

### A.4.1. CORE FILE OPERATIONS

**READ_FILE**: Reads specific files from the memory system with configurable character ranges (default: 5000-10000 characters). Parameters include `file_path`, `character_start`, and `character_end`. Used during in-session to access specific user model files or session data.

**SEARCH_FILE**: Performs BM25-based semantic search or string matching across the three-tier memory system. Parameters include `query`, `search_scope` (cleaned_sessions, session_analyses, user_profiles), `search_method` (bm25, string_match), `max_results`, `chunk_size`, and `latest_first`. Supports both exact substring matching and semantic ranking with English stemming.

**UPDATE**: Modifies JSON fields in the overall user model using dot notation paths. Parameters include `field_path`, `new_value`, `list_operation` (append, remove), `create_if_missing`, and `backup`. Supports list operations with duplicate prevention and automatic timestamping.

### A.4.2. MEMORY SYSTEM PROCESSING

**ANALYZE_SESSION**: Processes batches of raw session data to create session-based user models (Tier 2). Parameters include `user_id` and `session_batch`. Leverages LLM to extract user intent, emotional states, and message-level preferences using structured Pydantic models (see A.5 for examples).

**INITIALIZE_USER_PROFILE**: Aggregates session analyses to create or update overall user models (Tier 3). Parameters include `user_id`. Consolidates behavioral patterns across sessions into comprehensive user profiles with preference clusters and interaction style summaries (see A.5 for examples).

A.4.3. RESPONSE GENERATION ACTIONS

**GIVE_SUGGESTIONS**: Produces final in-session consultation responses containing personalized suggestions for the SWE agent. Parameters include `suggestions` and `confidence_score` (0-1 range). Returns structured `GenerateSuggestionsParams` objects with user modeling insights.

A.4.4. IMPLEMENTATION ARCHITECTURE

The action execution framework uses a workflow controller pattern with structured LLM calls, preset action sequences, and iterative refinement (maximum 3 iterations by default). All actions support comprehensive error handling, automatic retry mechanisms with exponential backoff, and validation through Pydantic schemas. The system includes monitoring capabilities with structured logging and metrics collection for debugging and optimization.

## A.5. Memory System JSON Examples

A.5.1. OVERALL USER MODEL EXAMPLE

*Listing 1.* Overall User Model Example

```
{
  "user_id": "dev_alice_2024",
  "profile_description": "Senior backend developer, prefers TypeScript",
  "interaction_style": {"verbosity": "concise", "question_timing": "upfront"},
  "coding_preferences": ["Always add type annotations", "Write tests first"],
  "session_summaries": [
    {"session_id": "2024-01-15_api_refactor", "tldr": "Refactored REST API"},
    {"session_id": "2024-01-20_auth_system", "tldr": "Implemented JWT auth"}
  ]
}
```

A.5.2. SESSION-BASED USER MODEL EXAMPLE

*Listing 2.* Session-based User Model Example

```
{
  "session_id": "2024-01-25_database_migration",
  "user_intent": "Migrate from MongoDB to PostgreSQL",
  "user_profile": "Backend developer, prefers step-by-step validation",
  "message_preferences": [
    {"message_id": 1, "user_message": "Help me migrate the user data",
     "inferred_constraints": ["preserve data integrity"],
     "preferred_approach": "incremental migration with validation"}
  ]
}
```

## A.6. Developer Profiles Breakdown

*Table 3.* 15 Developer Profiles in Stateful SWE Benchmark

| Profile ID | Interaction Style | Coding Preferences (sample) |
|---|---|---|
| P01 | Concise + Upfront + Short | Always use exact same branch name when updating... |
| P02 | Concise + Ongoing + Short | Use descriptive branch names like 'feature/user-auth'... |
| P03 | Verbose + Upfront + Verbose | Use develop branch as primary development branch... |
| P04 | Verbose + Ongoing + Verbose | Clean up merged branches regularly to maintain... |
| P05 | Verbose + Upfront + Short | Implement comprehensive test coverage: unit, integration... |
| P06 | Verbose + Ongoing + Short | Implement comprehensive test coverage: unit, integration... |
| P07 | Concise + Upfront + Verbose | Use rebasing over merging to maintain clean git history... |
| P08 | Concise + Ongoing + Verbose | Always use exact same branch name when updating... |
| P09 | Concise + Upfront + Short | Be comfortable with force push for updating existing PRs... |
| P10 | Concise + Ongoing + Short | Clean up merged branches regularly to maintain... |
| P11 | Verbose + Upfront + Verbose | Write descriptive commit messages explaining the 'why'... |
| P12 | Verbose + Ongoing + Verbose | Separate git push operations from PR/MR creation... |
| P13 | Verbose + Upfront + Short | Use develop branch as primary development branch... |
| P14 | Verbose + Ongoing + Short | Use develop branch as primary development branch... |
| P15 | Concise + Upfront + Verbose | Use rebasing over merging to maintain clean git history... |

## A.7. Prompt Templates

The ToM agent operates through five core Jinja2 templates that implement the prompting methodology described in Section 2:

### A.7.1. WAKE-TIME SUGGESTION TEMPLATE

*Listing 3.* give_suggestions.jinja2 (key components)

```
You are the ToM Agent expert in modeling user mental state and behavior.
Your job is to provide suggestions to the SWE agent based on user modeling.

Available Actions:
- SEARCH_FILE: Find relevant behavior patterns (BM25 search)
- READ_FILE: Read specific user model files
- GENERATE_SUGGESTIONS: Provide final recommendations (mandatory final action)

Special Cases: GitHub Issue Analysis, Empty instructions, Hard to recover scenarios
```

### A.7.2. SLEEP-TIME MEMORY UPDATE TEMPLATE

*Listing 4.* update_memory.jinja2 (key components)

```
You are a user modeling expert processing session files through three-tier memory.

Available Actions:
- UPDATE_JSON_FIELD: Update overall_user_model fields (append, remove operations)
- GENERATE_SLEEP_SUMMARY: Provide final summary (mandatory final action)

Key: Update UserProfile fields, include specific preferences, use [IMPORTANT] tags
```

### A.7.3. SESSION ANALYSIS TEMPLATE

*Listing 5.* session_analysis.jinja2 (excerpt)

```
Analyze this coding session to understand the user's behavior, intent, and preferences.

## Full Session Context:
{{ full_session_context }}

## Key User Messages (focus on these for analysis):
{{ key_user_messages }}
```

```
## Session Metadata:
- Session ID: {{ session_id }}
- Total messages: {{ total_messages }}
- Important user messages: {{ important_user_messages }}
```

### A.7.4. USER ANALYSIS TEMPLATE

*Listing 6.* user_analysis.jinja2 (excerpt)

```
Analyze these recent coding sessions to create a comprehensive user profile.

User ID: {{ user_id }}
Recent Sessions ({{ num_sessions }} sessions):
{{ sessions_text }}

Create a user analysis including: overall description, intent/emotion distributions, preferences
For the preferences, pay attention to different kinds of preferences:
- Interactional preferences: how users prefer to communicate with the SWE agent, concise vs verbose responses,
    upfront vs ongoing question timing, short vs long responses
- Coding preferences: TypeScript, React, Node.js, testing practices, etc.
- Other preferences: special requirements for the SWE agent
```

### A.7.5. MESSAGE CONDENSATION TEMPLATE

*Listing 7.* message_condensation.jinja2 (excerpt)

```
Please condense the following message to max {{ max_tokens }} tokens (do not exceed the limit, and do not add any
    extra information).
FOCUS: Keep the most important information that provides context for understanding a conversation.

Original message:
{{ content }}

Condensed version:
```

## A.8. Experimental Configuration

### A.8.1. MODEL SPECIFICATIONS

Our experiments use the following model configurations. The primary ToM agent model is **Claude Sonnet 4** (claude-sonnet-4-20250514), which provides the core user modeling capabilities. For baseline comparison, we use **Claude 3.7 Sonnet** (claude-3-7-sonnet-20241022). Additionally, we conduct multi-model evaluation using **GPT models** including gpt-5-nano-20241201, gpt-5-mini-20241201, and gpt-5-20241201.

### A.8.2. USER SIMULATOR SATISFACTION EVALUATION

User simulator satisfaction scores are computed through an automated evaluation pipeline using profile-conditioned user simulators powered by GPT-5. The evaluation process consists of three key components: (1) **Trajectory Analysis**: The complete agent-user interaction history, including user messages, agent responses, code changes, and final outputs, is formatted into a structured conversation flow for evaluation; (2) **Profile-Conditioned Assessment**: Each user simulator is instantiated with the specific developer profile used during the original interaction, ensuring consistent evaluation criteria based on the user's stated preferences for verbosity, question timing, and coding practices; (3) **Multi-Dimensional Scoring**: The simulator evaluates agent performance across five dimensions on a 1-5 scale: overall satisfaction, communication quality, problem-solving approach, efficiency, and user preference alignment.

**Quality Control**: We implemented preliminary validation by manually reviewing 30 randomly sampled satisfaction scores across different agent types for Claude 4. We found that the human evaluation and the user simulator have substantial correlation ($r = 0.86$, $p < 0.001$) for overall satisfaction scores. We also validated that satisfaction scores appropriately reflect profile-specific preferences by confirming that agents violating explicit user preferences (e.g., asking excessive questions to concise users) received correspondingly lower scores.

### A.8.3. EVALUATION FRAMEWORK

The multi-model comparison framework consists of three main components. The evaluation runner serves as the main orchestration system, supporting model filtering, sample size control, and parallel evaluation across different model

configurations. The core evaluation logic implements the clarity assessment framework for analyzing model performance across different conditions. Configuration management is handled through structured configuration objects that specify model parameters, API endpoints, and evaluation settings for each experimental condition.

### A.8.4. HYPERPARAMETERS

The experimental setup uses carefully tuned hyperparameters across different system components. For **memory retrieval**, we retrieve the top-k=3 most relevant sessions BM25 search. The **ToM action limit** restricts the agent to a maximum of 3 memory actions per consultation to balance thoroughness with efficiency. **Temperature settings** are configured as 0.1 for the ToM agent to ensure consistent user modeling outputs, and 0.7 for the SWE agent to maintain appropriate creativity in code generation.

### A.8.5. SWE AGENT PROMPT FOR BENCHMARK TASKS

The following prompt template is used for all agents (CodeAct, RAGCodeAct, and TomCodeAct) when working on benchmark tasks. The template provides the agent with context about the workspace, task requirements, and operational constraints:

*Listing 8.* SWE Agent Benchmark Task Prompt

```
<uploaded_files>
/workspace/{{ workspace_dir_name }}
</uploaded_files>

Can you help me implement the necessary changes to the repository so that
the requirements specified in the <issue_description> are met?
Relevant python code files are in the directory {{ workspace_dir_name }}.
DON'T modify the testing logic or any of the tests in any way!
Also the development Python environment is already set up for you (i.e., all
dependencies already installed), so you don't need to install other packages.
You are encouraged to use non tool_calls actions to engage with the user/me,
including providing progress reports, answering questions, asking for
clarification, etc. Once you issue the finish action, it means you are
confident that you have solved the issue. Any time before that, you will have
the opportunity to communicate with the user/me to resolve the issue better.

<issue_description>
{{ instance.problem_statement }}
</issue_description>
```

This prompt establishes the task context, workspace boundaries, and interaction expectations. The `workspace_dir_name` and `instance.problem_statement` variables are populated with instance-specific information from the benchmark dataset.

### A.9. Data Formats and JSON Schemas

### A.9.1. USER PROFILE SCHEMA

*Listing 9.* Overall User Model Schema

```
{
  "user_id": "dev_alice_2024",
  "profile_description": "Senior backend developer, prefers TypeScript",
  "interaction_style": {"verbosity": "concise", "question_timing": "upfront"},
  "coding_preferences": ["Always add type annotations", "Write tests first"],
  "session_summaries": [
    {"session_id": "2024-01-15_api_refactor", "tldr": "Refactored REST API"}
  ]
}
```

### A.9.2. SESSION MODEL SCHEMA

*Listing 10.* Session-based User Model Schema

```
{
  "session_id": "2024-01-25_database_migration",
  "user_intent": "Migrate from MongoDB to PostgreSQL",
  "user_profile": "Backend developer, prefers step-by-step validation",
```

```
  "message_preferences": [
    {"message_id": 1, "user_message": "Help me migrate the user data",
     "inferred_constraints": ["preserve data integrity"],
     "preferred_approach": "incremental migration with validation"}
  ]
}
```

## A.10. Simulated User Judgement Prompt

Here's our simulated user judgement prompt. Given the user profile, full issue, underspecified issue, and the full agent trajectory, we ask the LLM to evaluate the agent from a user's perspective on these dimensions (1-5 scale):

1. **OVERALL_SATISFACTION**: Based on the user profile, would a user be satisfied with this interaction?

2. **COMMUNICATION_QUALITY**: How well did the agent communicate? Did the agent ask appropriate clarifying questions only when necessary?

3. **PROBLEM_SOLVING_APPROACH**: Was the approach systematic and effective?

4. **EFFICIENCY**: Did the agent work efficiently without violating the user profile's preferences?

5. **USER_PREFERENCE_ALIGNMENT**: Did the agent respect user preferences and context?

An overall concrete example of simulated user judgement includes:

*Listing 11.* Simulated User Judgement Example

```
{
  "scores": {
    "overall_satisfaction": 3.3,
    "communication_quality": 2.9,
    "problem_solving_approach": 3.8,
    "efficiency": 3.2,
    "user_preference_alignment": 3.0
  },
  "explanation": "Good attempt and technically plausible approach, but the main
    issue persists and tests still fail. Communication lacked up-front
    clarification and verification against user priorities (clean build, fix
    failing tests first). Result feels incomplete and risky without broader
    validation.",
  "detailed_feedback": {
    "strengths": "Systematically located the relevant code paths, built a
      minimal repro, and implemented a focused change (ref-specific,
      fuzzy-match prioritization, clearer warnings). Kept tests untouched and
      produced a clear commit message. Verified by building docs multiple
      times.",
    "weaknesses": "Did not ensure tests pass or add unit coverage for the new
      behavior. The root issue remains unresolved per the evaluation, and
      there was no rollback/mitigation plan. Risk of global behavior change
      without validating Sphinx's existing test suite. Communication of
      trade-offs and scope was limited. No up-front clarifying questions to
      confirm expectations. Potentially broad patch with unknown side effects.",
    "user_experience": "Still seeing noisy builds and at least one failing
      test. Improvements seem partial and unverified across the codebase,
      leaving me uncertain about stability and correctness. Appreciated the
      attempt, but I still need a clean, reliable result.",
    "question_asking_behavior": "No clarifying questions asked at the
      beginning to confirm desired behavior or constraints. While they avoided
      mid-task questions, they also missed the chance to align early on
      requirements (e.g., strict test-passing policy)."
  }
}
```

## A.11. Complete Developer Profile Example

Here's a concrete example of a complete developer profile used in our experiments:

*Listing 12.* Complete Developer Profile Example

```
You are roleplaying as a software developer with these characteristics:
```

```
INTERACTION STYLE:
- You prefer brief, to-the-point responses. You get impatient with long
  explanations and often say things like 'keep it short' or 'just the
  essentials please'.
- You prefer to ask all your clarifying questions at the beginning before
  any work starts. You like to understand the full scope upfront. You won't
  answer any questions if the agent ask questions in the middle or at the
  end of the work.
- You respond concisely and to the point. Your answers for the SWE agent are
  usually under 15 words. When facing multiple questions, you will usually
  only answer the first question and ignore the rest.

CODING STANDARDS:
- You have specific coding preferences: Always use exact same branch name
  when updating existing work; Implement standardized error handling patterns
  across entire codebase; Always create proper database migration scripts for
  schema changes; Choose Biome over ESLint+Prettier for JavaScript/TypeScript
  linting and formatting; Use httpx over requests library for Python HTTP
  clients with proper async support; Write minimal but meaningful code
  comments, prefer self-documenting code structure; Write descriptive commit
  messages explaining the 'why' not just 'what'; Use environment variables
  for configuration over hardcoded values; Follow PR templates when available
  rather than ad-hoc descriptions; Write comprehensive API documentation with
  examples and complete response format specifications

Respond naturally as this type of user would, incorporating these preferences
into your messages. Be authentic to this persona while working with the SWE
agent.
```

*Listing 13.* Complete Developer Profile Example 2

```
You are roleplaying as a software developer with these characteristics:

INTERACTION STYLE:
- You appreciate detailed explanations and comprehensive responses. You often
  ask for more details and thank the agent for thorough breakdowns.
- You're comfortable with questions being asked throughout the process as
  they arise. You prefer iterative clarification.
- You could provide more detailed answers for the SWE agent. You are willing
  to answer more than one question from the SWE agent.

CODING STANDARDS:
- You have specific coding preferences: Use descriptive branch names like
  'feature/user-auth' or 'DAISY-1046'; Create tests before or alongside
  implementation, not as afterthought; Integrate automated linting (ESLint,
  Biome, Ruff) in development workflow; Use async/await patterns consistently
  for all concurrent operations; Implement standardized error handling
  patterns across entire codebase; Implement JWT-based authentication with
  proper OIDC integration and session management; Build React applications
  with TypeScript and proper component organization patterns

Respond naturally as this type of user would, incorporating these preferences
into your messages. Be authentic to this persona while working with the SWE
agent.
```

*Listing 14.* Complete Developer Profile Example 3

```
You are roleplaying as a software developer with these characteristics:

INTERACTION STYLE:
- You prefer brief, to-the-point responses. You get impatient with long
  explanations and often say things like 'keep it short' or 'just the
  essentials please'.
- You're comfortable with questions being asked throughout the process as
  they arise. You prefer iterative clarification.
- You could provide more detailed answers for the SWE agent. You are willing
  to answer more than one question from the SWE agent.

CODING STANDARDS:
- You have specific coding preferences: Handle pushing changes to multiple
  repositories simultaneously when needed; Enforce strict TypeScript
  compliance and comprehensive type checking; Use PostgreSQL over MySQL with
  proper ORM patterns (SQLAlchemy, Django ORM); Create interactive
  documentation with collapsible sections and expandable code blocks; Include
  cross-references and links between related documentation sections

Respond naturally as this type of user would, incorporating these preferences
into your messages. Be authentic to this persona while working with the SWE
agent.
```

## A.12. Coding Session Summarization Prompt

The categories of user preferences and interaction patterns presented in this paper are summarized from 209 coding sessions collected during our user study with 17 professional developers. We did not provide users with predefined tasks; instead, developers chose their own tasks aligning with their daily work. To extract structured insights from these sessions, we used the following LLM-based summarization prompt:

*Listing 15.* Coding Session Message Analysis Prompt

```
{
  "type": "OBJECT",
  "properties": {
    "message_content": {
      "type": "STRING",
      "description": "The content of the user message (Try to keep the
        original message as much as possible unless it's too long or contains
        too much user copy-pasted content; in a nutshell, try to preserve what
        the user actually said and include details if possible)"
    },
    "emotions": {
      "type": "STRING",
      "description": "A description of the emotional states detected in the
        message. Choose from: frustrated, confused, confident, urgent,
        exploratory, focused, overwhelmed, excited, cautious, neutral."
    },
    "preference": {
      "type": "STRING",
      "description": "A description of the preferences that the user has. Be
        specific about the preferences and extract in a way that could be
        useful for helping better understand the user intents in the future."
    }
  },
  "propertyOrdering": [
    "message_content",
    "emotions",
    "preference"
  ]
}
```

This structured analysis enabled us to systematically identify patterns across interaction styles, coding preferences, and communication behaviors from real-world developer sessions.

