# OpenReview forum: "TOM-SWE: User Mental Modeling For Software Engineering Agents"
_ICML.cc/2026/Conference — ICML 2026 regular_

### Official Review · Reviewer_DfET · 2026-03-07

**Soundness:** 4
**Presentation:** 4
**Significance:** 4
**Originality:** 3
**Overall Recommendation:** 5
**Confidence:** 5

**Summary:**

The paper proposes the ToM-SWE algorithm as a plug-in to traditional SWE agents that adds a separate agent that collects user preferences, updates them and makes the main agent consult them while running. The experiments show the value of the addition in terms of the final score. A decently sized human study has been conducted to analyze the user satisfaction levels.

**Compliance With Llm Reviewing Policy:**

Affirmed.

**Ethical Review Concerns:**

Mostly the concerns are around the impact of preference profiling on the human SWE users - personal mental impact. Another concern is about the privacy of the collected user profiles, which are inevitable in the commercial setting but not an issue within the scope of the study, since explicit consents were taken.

**Ethics Expertise Needed:**

["Privacy and Security (e.g., personally identifiable information)"]

**Key Questions For Authors:**

Surprisingly, none.

**Limitations:**

The claimed results are specific to SWE, but not the generic HMI.

**Strengths And Weaknesses:**

# Strengths

[1] The motivation for the study is clearly laid out.

[2] The level of score improvement claimed on Ambiguous SWE-bench and the self-designed Stateful SWE-bench are significant.

[3] The feedback study from professional software developers is valuable.

[4] The architecture of SWE + ToM agents is clear.

[5] Figure 4 displays the convincing superiority of TomCodeActAgent relative to CodeAct and RAG-enhanced CodeAct. The p-value measurement at line 295 is legit.

[6] Section 6 discloses enough technical details about the human SWE study and is well designed.

[7] Section “Context Specificity Spectrum” is insightful.

[8] Listings 12-14 of the collected user preferences are insightful and highly appreciated.

# Weaknesses

[1] From the ethical perspective a user must be able to query their personal profile that the system collected on them. The impact of finding out what AI thinks about a person-user has not been studied. I can imagine a scenario where the user’s personal profile contains what may be considered offensive records like “repeatedly falls into fallacy X” or “stubbornly insists me to follow anti-pattern Y” and negatively affect their mental state. As a follow-up study I’d suggest assessing the level of emotional damage or lack thereof in a setup when human developers are presented with the collected personal profiling document.

[2] Cursor’s continual learning plug-in already implements collection of user preferences from all historic chats. https://cursor.com/marketplace/cursor/continual-learning

[3] The term “mental state” is chosen incorrectly and should be replaced with “user preferences”.

[4] It is not clear why a duet of DWE and ToM agents is needed and why a single agent can’t do both functions. Line 200 states this but no evidence has been provided to support it.

---

> ### Author Rebuttal · Authors · 2026-03-30
>
> We thank the reviewer for the detailed assessment and for highlighting the "convincing superiority" shown in Figure 4 and the "insightful" context specificity analysis.
>
> > **W1: "From the ethical perspective a user must be able to query their personal profile... I can imagine a scenario where the user's personal profile contains what may be considered offensive records like 'repeatedly falls into fallacy X' or 'stubbornly insists me to follow anti-pattern Y.'"**
>
>  This is an important concern. We note that our profiles are designed around factual *preferences* (e.g., "prefers concise responses," "uses httpx over requests"), not evaluative judgments about user competence, see Appendix A.6 and A.9 for full profile examples. However, we agree that profile transparency is essential and will add to the revised paper: (1) users can view and edit their profiles at any time, and (2) the reviewer's excellent suggestion for a follow-up study "assessing the level of emotional damage or lack thereof" when developers are shown their profiles will be noted as important future work in our ethics discussion.
>
> > **W2: "Cursor's continual learning plug-in already implements collection of user preferences from all historic chats."**
>
> Thank you for pointing this out! Cursor’s continual learning plugin came out on February 20th, 2026, after this paper was authored, arXived, and submitted to ICML. Hopefully our paper was useful to the authors of that plugin at Cursor!
>
>
> That being said, there are important differences: (1) Cursor's system passively stores user rules within a single tool's ecosystem; our ToM agent performs active reasoning about user intent during each interaction; (2) Cursor does not explicitly mention separating user modeling from coding . It is likely a single-agent system, which our experiments show underperforms; (3) no systematic evaluation framework exists for Cursor's feature, our Stateful SWE-bench provides the first such benchmark. We will cite Cursor's continual learning and clarify these distinctions in the revision.
>
> > **W3: "The term 'mental state' is chosen incorrectly and should be replaced with 'user preferences.'"**
>
>  "User preferences" captures part of our memory system. However, the ToM agent also infers in-session intent, communication expectations, and emotional cues, for example, detecting when a user is becoming frustrated from terse responses and adjusting the SWE agent's behavior. These go beyond static preferences. We can clarify the full extent of what we are modeling in our paper.
>
>
> > **W4: "It is not clear why a duet of SWE and ToM agents is needed and why a single agent can't do both functions. Line 200 states this but no evidence has been provided to support it."**
>
> Please note that in 323-329, we gave the single-agent CodeAct full access to all three memory tiers (including the Tier-3 user summary) on 100 Stateful SWE-bench instances with Claude Sonnet 4.  And we showed that a single agent burdened with both tasks cannot leverage user context as effectively as a dedicated ToM agent. We will make this evidence more prominent in the revised text to ensure it is not overlooked.

---

> > ### Author Rebuttal · Reviewer_DfET · 2026-04-01
> >
> > Thank you for your answers. Great job!

---

### Official Review · Reviewer_7Kno · 2026-03-08

**Soundness:** 4
**Presentation:** 3
**Significance:** 3
**Originality:** 3
**Overall Recommendation:** 4
**Confidence:** 4

**Summary:**

This paper proposes a dual-agent framework that introduces a ToM agent alongside a primary coding agent. The ToM agent is specifically designed to model and maintain user memory and preferences. By doing so, the framework achieves improved performance and better user alignment in scenarios involving ambiguous instructions and personalised programming tasks.

**Compliance With Llm Reviewing Policy:**

Affirmed.

**Final Justification:**

In my final assessment, the rebuttal addressed most concerns. From a methodological perspective, user modelling is broadly applicable to many agent systems, regardless of whether the users are programmers, designers, or patients. The underlying approaches also appear similar across domains, typically involving domain-specific adaptations built on top of RAG, which limits the perceived originality.

Nevertheless, the paper’s contributions extend beyond the method itself. The dataset and the real-world observational study strengthen the work in terms of significance and practical relevance. The rebuttal helped clarify these strengths. I'd like to increase the originality score to 3 while keeping my overall assessment unchanged.

**Key Questions For Authors:**

Please see the weakness above.

**Limitations:**

yes

**Strengths And Weaknesses:**

Strengths:
- The paper addresses a highly practical pain point in real-world software engineering by tackling the difficulty coding agents face with underspecified and context-dependent instructions.
- Conducting a three-week "in-the-wild" observational study with professional developers working on their own unscripted daily tasks is a major strength. This provides compelling evidence of the system's practical utility beyond simulated lab environments.
- The dual-agent architecture separating the SWE agent and the ToM agent is well-justified. Furthermore, the lightweight ToM agent operates with high cost-efficiency, substantially improving performance without introducing significant computational or financial overhead.
- The authors make a solid contribution to the community by introducing the Stateful SWE benchmark. This novel dataset is highly valuable for evaluating long-horizon interactions, as it allows agents to leverage realistic conversation histories across multiple sessions.

Weaknesses:
- The ToM framing is overstated. Rather than genuinely modelling a user's BDI, the system essentially implements conventional multi-level memory and preference tracking.
- The framework lacks architectural novelty. Cross-session hierarchical memory and retrieval are already standard in personalised agents, making this paper more like a solid industrial application of best practices in AI coding.

---

> ### Author Rebuttal · Authors · 2026-03-30
>
> We thank the reviewer for recognizing the practical motivation, the dual-agent justification, and Stateful SWE-bench as "highly valuable."
>
> > **W1: "The ToM framing is overstated. Rather than genuinely modelling a user's BDI, the system essentially implements conventional multi-level memory and preference tracking."**
>
> We do not claim to implement a full BDI architecture. Our use of "Theory of Mind" follows the cognitive science definition: the capacity to attribute mental states to others — inferring what a developer intends, prefers, and expects from observed behavior.
>
> The critical distinction from "conventional memory and preference tracking" is demonstrated empirically: RAGCodeAct has access to the same historical data and retrieves relevant information via BM25, yet consistently underperforms. The ToM agent's value lies in *interpreting* interactions to build a coherent user model, not just preferences, but also personality, emotion, and etc. those cognitive activities across coding sessions.
>
> We revisited the current paper wording and believe it is already reasonably clear with respect to this, but if you have suggestions on how the paper could be further clarified we would be happy to entertain them.
>
>
> > **W2: "The framework lacks architectural novelty. Cross-session hierarchical memory and retrieval are already standard in personalised agents, making this paper more like a solid industrial application of best practices."**
>
> We respectfully note that the contribution is not the individual architectural components. The novelty lies in: (1) the *problem formulation*, framing SWE agent personalization as user mental modeling, which is new in this domain; (2) the *dual-agent paradigm*, demonstrating empirically that separating user modeling from coding is essential. (3) *Stateful SWE-bench*, the first benchmark for cross-session user modeling in SWE agents, which the reviewer themselves calls "highly valuable"; and (4) *validation at scale*, 500 instances × 3 models × 3 agents + a real developers three-week long field study. As far as we know, there are no industrial best practices with respect to any of these contributions.

---

> > ### Author Rebuttal · Reviewer_7Kno · 2026-04-02
> >
> > My point is that, at the level of concrete methodology, most agent systems would benefit from user modelling, regardless of whether the user is a programmer, designer, or patient. Moreover, the approaches to user modelling are largely similar, typically involving domain-specific adaptations built on top of RAG. That said, I do acknowledge the value of this work, which lies not only in the proposed method but also in the dataset and the real-world observational study. I will increase the originality score to 3.

---

### Official Review · Reviewer_MvjS · 2026-03-12

**Soundness:** 3
**Presentation:** 3
**Significance:** 2
**Originality:** 3
**Overall Recommendation:** 4
**Confidence:** 4

**Summary:**

This paper introduces ToM-SWE, a dual-agent framework that adds a dedicated theory-of-mind (ToM) agent alongside a standard SWE coding agent. The ToM agent tracks user preferences, intent, and interaction style through a three-tier hierarchical memory system and provides suggestions to the SWE agent when needed. The paper also contributes Stateful SWE-bench, a new benchmark that brings personalized user profiles and cross-session interaction histories into SWE evaluation for the first time. Experiments span two benchmarks, three base models, and a three-week field study with 17 professional developers. The main takeaway is that explicit user modeling leads to large gains on the stateful benchmark and that developers in the wild find ToM suggestions useful most of the time.

**Compliance With Llm Reviewing Policy:**

Affirmed.

**Key Questions For Authors:**

1. What if you just give the SWE agent the Tier-3 user summary directly, no ToM agent reasoning in the loop? That would tell you how much of the gain comes from having structured user info versus having an agent actively thinking about it.
2. What if you swap the vague instructions back to the full original SWE-bench descriptions but keep the personalized simulator and history? That would help separate "figuring out what the user wants" from "adapting to how the user wants it done."

**Limitations:**

1. The performance gap on Stateful SWE-bench is suspicious. CodeAct goes from 68% on original SWE-bench down to 18% on the stateful version — that's a cliff. So when TomCodeAct hits 59.7%, how much of that is actually understanding user preferences, and how much is just the ToM agent helping recover task information from the history that the baseline couldn't get at? The paper doesn't really pull these two things apart.
2. The whole setup feels hard to scale. You need to collect real sessions, build profiles, run GPT-5 to align messages, maintain a three-tier memory system per user — that's a lot of moving parts. The paper doesn't discuss what happens when you have hundreds or thousands of users, or when user preferences shift over time. It works as a research prototype, but the path to something practical at scale isn't clear.
3. The human study has no control group. All 17 developers used the ToM version — nobody used a plain baseline. So when you see 86% acceptance, you don't really know what to compare it to. Maybe a simpler system would also get high acceptance. And "Accept" could just mean the suggestion was harmless, not that it actually helped.

**Strengths And Weaknesses:**

1. The problem is realistic and well-motivated. Current coding agents treat every session as new, which is not efficient enough for long-term users. The paper identifies a meaningful gap and makes a clear case for why user modeling matters in software development.
2. Good design for two agents separately. The single-agent baseline that tries to do both only hits about 27% on the stateful benchmark, and RAG-based retrieval actually hurts performance on weaker models. These results make the dual-agent design feel necessary rather than arbitrary.
3. Stateful SWE-bench is a nice contribution on its own. They actually went and collected real sessions, built diverse profiles, used GPT-5 to align messages to persona styles, and made user simulators that behave differently per profile — like concise users just refusing to answer if you bombard them with questions. The 15 profiles are fleshed out well in the appendix, and you can tell real effort went into making this feel grounded.
4. Contain the cost analysis. This makes it easy to see that even the cheapest ToM model gives you a meaningful boost for basically nothing per session. This is the kind of thing that actually helps someone decide whether to use this in practice.

---

> ### Author Rebuttal · Authors · 2026-03-30
>
> We thank the reviewer for the thoughtful questions and for recognizing the dual-agent design as "necessary rather than arbitrary."
>
> > **Q1: "What if you just give the SWE agent the Tier-3 user summary directly, no ToM agent reasoning in the loop?"**
>
> Please note that in 323-329, we gave the single-agent CodeAct full access to all three memory tiers (including the Tier-3 user summary) on 100 Stateful SWE-bench instances with Claude Sonnet 4.  This setup achieved only a 27.17% resolution rate on the stateful SWE bench, falling significantly short of ToMCodeAct’s 59.7%.
>
> > **Q2: "What if you swap the vague instructions back to the full original SWE-bench descriptions but keep the personalized simulator and history?"**
>
> Our Ambiguous SWE-bench experiment partially isolates this: it uses underspecified instructions requiring clarification, but without personalized profiles or history. Swapping the vague instructions back to full original SWE-bench descriptions would require no interaction between the simulated user and coding agents to achieve a high success rate. We believe that this would not meaningfully measure either the “what” or “how” that was pointed out in the review comment, but if we are misunderstanding something we’d be happy for any clarification.
>
> > **L1: "CodeAct goes from 68% on original SWE-bench down to 18% on the stateful version — that's a cliff. How much of that is actually understanding user preferences, and how much is just the ToM agent helping recover task information from the history?"**
>
> The cliff exists by design: Stateful SWE-bench replaces precise issue descriptions with vague user instructions, requiring agents to interact with users to disambiguate. CodeAct drops to 18% because it cannot effectively recover task intent through interaction, not because the underlying tasks are harder (they are the same SWE-bench issues).
>
> TomCodeAct's gains come from both capabilities, and they are entangled by design: effective disambiguation *requires* understanding how to interact with this particular user. A concise user who prefers upfront questions needs a fundamentally different interaction strategy than a verbose user comfortable with ongoing clarification. Our two benchmarks partially disentangle these: Ambiguous SWE-bench isolates disambiguation without personalization, while Stateful SWE-bench (which is more ambiguous), further adds personalization.
>
>
> > **L2: "The whole setup feels hard to scale... The paper doesn't discuss what happens when you have hundreds or thousands of users, or when user preferences shift over time."**
>
> We agree scalability deserves more discussion and we will add this to the revised draft. That said, the architecture supports scaling: (1) memory is per-user and Tier 3 compresses all history into a single profile, so storage grows sublinearly; (2) the ToM agent adds only $0.02–$0.17/session (Section 5.3); (3) after-session processing runs asynchronously. Preference drift is handled by the incremental UPDATE action, which modifies the user model after each new session. We acknowledge that scaling to thousands of users and studying long-term drift is important future work, but the current design already supports it architecturally.
>
> > **L3: "The human study has no control group. All 17 developers used the ToM version — nobody used a plain baseline."**
>
> On one hand, we do agree that the human study could have benefitted from a control group, but on the other hand we also argue that including any human study at all goes significantly beyond the standard level of validation for an ICML paper. We chose to not have a control group, as collecting natural human feedback is quite time consuming, and as our observational field study, we wanted to collect as much data as we could about the behavior of the ToM-SWE agent to be able to analyze the results of our proposed method. We hope that this, supplemented by the offline benchmarks (Section 5) providing the controlled comparison, provides a sufficient holistic view of ToM-SWE’s performance. We will clarify this complementary design more explicitly in the revised paper.
>
>
> The 86.2% acceptance rate is intrinsically meaningful as developers could freely reject at zero cost, with no social pressure or experimenter observation (especially the acceptance is about the suggestion from the ToM agent, which would not influence the task execution of coding agents if the suggestion gets rejected). Tasks were highly diversified across different codebases with minimal overlap between users, ruling out a narrow task distribution.

---

> > ### Author Rebuttal · Reviewer_MvjS · 2026-04-03
> >
> > The rebuttal addresses my questions. I will keep my positive attitude for this paper.

---

### Official Review · Reviewer_BBVj · 2026-03-24

**Soundness:** 2
**Presentation:** 3
**Significance:** 4
**Originality:** 4
**Overall Recommendation:** 5
**Confidence:** 4

**Summary:**

This work proposes a dual-agent architecture for Software Engineering agents, ToM-SWE, that pairs a primary coding agent with a Theory of Mind (ToM) agent, which iteratively maps the user's preferences over the course of multiple sessions. ToM Agent's main task is to maintain a hierarchical memory of user preferences and intents, retrieve the memories, and help the SWE-Agent to complete ambiguous user queries successfully while respecting the user's preferences.

Authors also introduce a new variant of SWE-Bench, called Stateful SWE-Bench, which preserves the past interaction histories for developers and is built to evaluate the user preference modeling for Software Engineering Agents. Stateful SWE-Bench is constructed by first collecting human-agent interactive sessions, building user profiles from the recorded sessions, and then pairing them with prior SWE-Bench tasks. The crucial difference is that Stateful SWE-Bench is an interactive benchmark, as opposed to SWE-Bench, and requires a User to interact with the agent through multiple software engineering sessions.
Experimental analysis is conducted using SWE-Bench, Ambiguous SWE-Bench, and Stateful SWE-Bench benchmarks. Authors evaluate three agents: CodeAct from OpenHands, RAGCodeAct, a modified version of CodeAct that manages both coding and user modeling, and the proposed ToM-SWE agent.

Experiments show that the ToM-SWE agent scores high on user satisfaction scores, both using user simulators and in-the-wild human study.

**Compliance With Llm Reviewing Policy:**

Affirmed.

**Key Questions For Authors:**

1. Since the ToM-SWE agent iteratively builds a user profile across sessions, it would be interesting to see a graph of satisfaction rate vs number of conversations. Can authors provide that?
2. Does the RAGCodeAct agent use the same hierarchical memory system that the ToM-SWE agent uses? If not, then there is a strict need to ablate the memory structure to understand if the gains are coming from the inclusion of a structured memory module or the separation of the two agents.
3. How are the 500 SWE-Bench instances paired with the 15 profiles? Is each instance paired with all 15 profiles? Apologies if this is already mentioned in the paper, and I have missed it.
4. If I am not wrong, the ToM Agent is always set to Claude Sonnet 4. Considering that the ToM agent is one of the main contributions of the work, it would be advisable to provide a comparison of models as a theory of mind agent.
5. In Section A.8.2, what is the inter-annotator agreement between human annotators?

**Limitations:**

yes

**Strengths And Weaknesses:**

## Strengths

- The problem of modeling user intent in software engineering tasks is well-motivated and timely.
- A two-agent design for modeling user intents and preferences is also well-motivated, even if not all the components are ablated.
- Stateful SWE-Bench fills an important gap in the literature. The benchmark appears well-constructed, featuring multiple developer profiles and paired user simulators.
- Experimental analyses are generally well done with only a few gaps.
- The paper is generally very well-written and easy to understand.

## Weaknesses

1. Authors use GPT-5 as both a user simulator and an LLM as a judge to obtain the satisfaction scores. This may bias the user simulator-based satisfaction scores. Moreover, human alignment verification is only conducted using 30 instances out of 500, which may not be enough to establish the alignment between user simulator scores and humans. Authors should also ask human annotators to rate the "perceived" satisfaction based on the trajectories.
2. Baseline selection for this work is a bit weak. Using CodeAct as a baseline for Stateful SWE-Bench is unfair, as it was never designed to carry over context. Ideally, authors should have validated the performance of the ToM-SWE agent against Agentic Memory systems. In my view, ToM agents act as a specialized case of agent memory.
3. The Human study is only conducted using the ToM-SWE agent. It is unclear if humans will have a similar satisfaction rate using a memory system or an RAGCodeAct baseline that users establish. While 86% satisfaction rate is impressive, without any control group or established baseline, it is hard to make a judgment based on this number.
4. Authors don't provide a satisfaction rate breakdown against the developer profile. That would be useful information. It is important to figure out if the ToM-SWE operates well with all types of developer profiles or the satisfaction rates driven by a specific profile.
5. The authors have designed a hierarchical memory system, but they don't provide any ablation of the memory system.

---

> ### Author Rebuttal · Authors · 2026-03-30
>
> We thank the reviewer for recognizing the significance (4/4) and originality (4/4) of our work.
>
> ### W1:
>
> We acknowledge the concern about potential self-preference bias when using the same model family. However, the simulator and judge serve separate roles with distinct prompts. To directly address the bias concern, we re-ran the satisfaction evaluation using Gemini-2.5-flash as an independent judge (100 sampled instances per agent, same evaluation prompt):
>
> | Agent | GPT-5 (Overall Sat.) | Gemini (Overall Sat.) | Gemini (Comm.) | Gemini (Pref. Align.) |
> |-------|------|------|------|------|
> | CodeAct | 2.57 | 3.73 | 1.53 | 2.81 |
> | +RAG | 3.09 | 3.85 | 2.39 | 3.09 |
> | +ToM | 3.62 | 3.94 | 3.16 | 3.48 |
>
> Gemini preserves the same ranking (TomCodeAct > RAGCodeAct > CodeAct).
>
> Our human-LLM alignment achieves a Pearson correlation of r=0.86 (t=8.92, p=1.13×10⁻⁹), with 95% CI [0.724, 0.932], demonstrating statistically significant agreement. We will report these significance tests more prominently in the revised paper.
>
> The 30 human ratings were produced by annotators reviewing complete agent-user interaction trajectories and independently scoring perceived satisfaction. We will make this connection clearer in the revised paper.
>
> ### W2:
>
> We agree that a simple CodeAct agent baseline is insufficient, which is why we also included the RAGCodeAct agent as a memory baseline. The fact that ToM-SWE outperforms RAGCodeAct indicates that having the same information performs much worse than *understanding the user* from that information. We will strengthen the baseline discussion in the revised paper to make this distinction clearer.
>
>
> ### W3:
>
> On one hand, we do agree that the human study could have benefitted from a control group, but on the other hand we also argue that including any human study at all goes significantly beyond the standard level of validation for an ICML paper. We chose to not have a control group, as collecting natural human feedback is quite time consuming, and as our observational field study, we wanted to collect as much data as we could about the behavior of the ToM-SWE agent to be able to analyze the results of our proposed method. We hope that this, supplemented by the offline benchmarks (Section 5) providing the controlled comparison, provides a sufficient holistic view of ToM-SWE’s performance. We will clarify this complementary design more explicitly in the revised paper.
>
> ### W4
>
> Thank you for the suggestion! Based on this, we performed a breakdown on all 500 Stateful SWE-bench instances:
>
> | Dimension | CodeAct | RAGCodeAct | TomCodeAct | Δ (Tom−Code) |
> |-----------|---------|------------|------------|--------------|
> | Concise users | 2.45 ± 0.90 | 3.01 ± 1.06 | 3.54 ± 0.87 | **+1.08** |
> | Verbose users | 2.67 ± 0.96 | 3.16 ± 0.95 | 3.70 ± 0.75 | **+1.03** |
> | Upfront questions | 2.51 ± 0.91 | 3.07 ± 0.99 | 3.64 ± 0.83 | **+1.12** |
> | Ongoing questions | 2.63 ± 0.96 | 3.10 ± 1.03 | 3.61 ± 0.80 | **+0.98** |
> | Short responses | 2.49 ± 0.92 | 2.96 ± 1.04 | 3.55 ± 0.86 | **+1.06** |
> | Verbose responses | 2.65 ± 0.96 | 3.23 ± 0.95 | 3.71 ± 0.76 | **+1.05** |
>
>
> ### W5
>
> The RAGCodeAct vs TomCodeAct comparison is a partial ablation: same data source (Tier 1), different processing depth. RAGCodeAct (Tier 1 only): 3.09 satisfaction; TomCodeAct (full three-tier): 3.62 satisfaction. Additionally (line 323-329), we test a single-agent baseline that has access to all the tiers of memory. Claude Sonnet 4 achieved a 27.17% resolution rate on the stateful SWE bench in this setup.
>
> ### Q1:
>
> Thank you for this suggestion! We will include such a graph in our updated draft.
>
> ### Q2:
>
> Thank you for the suggestion! To isolate the impact of the architecture, we tested a single-agent baseline equipped with all three tiers of memory, tasking it with managing both coding tasks and ToM-related contexts. In preliminary experiments (N=100 with Claude 4), this setup achieved only a 27.17% resolution rate on the stateful SWE bench. In our preliminary experiments, we found simply plugging Tiers 2 and 3 into RAGCodeAct is mechanistically ineffective. RAGCodeAct relies on query-based retrieval. Because an immediate task query is often lexically unrelated to underlying user preferences or historical interaction patterns, a standard RAG agent struggles to retrieve the relevant Tier 2 and Tier 3 contexts in the first place.
>
> ### Q3:
>
> Each instance is paired with exactly one profile via randomized assignment (`assignment_seed`).
>
> ### Q4:
>
> This is reported in Section 5.3: we tested five models as ToM base — GPT-5 nano (38.0%), GPT-5 mini, GPT-5, Claude 3.7, Claude 4 (59.7%). Even GPT-5 nano at $0.02/session substantially improves over no-ToM (18.1%).
>
> ### Q5:
>
> The Cohen’s Kappa between human annotators is 0.78, representing a relatively high agreement.

---

> > ### Author Rebuttal · Reviewer_BBVj · 2026-04-04
> >
> > Thank you for providing a detailed response, and apologies for the delayed response from my side. I believe all of my concerns are adequately addressed, except the baseline.
> >
> > While I agree with the authors that RAGCodeAgent is a useful baseline, and it does provide evidence that _understanding_ the user's preferences is important. At the same time, I will argue that the understanding comes from the memory structures and distillation of _raw_ notes and effectively utilizing them. Prior work has studied efficient memory architectures that essentially build a user preference model, albeit for different applications. One such work is GUM [1].
> > However, authors don't even cite this work. I believe comparison with such baselines is quite important for this work.
> >
> > I will keep my score, as I do believe this work should be accepted, and I already voted for acceptance. Authors should try their best to include GUM and other relevant works as baselines in their updated draft.
> >
> > [1] Shaikh, O., Sapkota, S., Rizvi, S., Horvitz, E., Park, J. S., Yang, D., & Bernstein, M. S. (2025, September). Creating general user models from computer use. In Proceedings of the 38th Annual ACM Symposium on User Interface Software and Technology (pp. 1-23).

---

> > > ### Author Response · Authors · 2026-04-04
> > >
> > > Thank you for pointing us to GUM (Shaikh et al., 2025). We will cite and discuss it in our revised draft.
> > >
> > > GUM's propose-retrieve-revise architecture for building user models from observations is indeed conceptually related to our ToM agent's approach of distilling user preferences into structured memory. The key difference lies in the interaction modality and purpose: GUM passively observes computer use (screenshots, clicks) to build a general-purpose user profile, whereas our ToM agent actively models preferences through conversational interaction within a specific task context, enabling it to resolve ambiguity in real time.

---

### Decision · Program_Chairs · 2026-04-30

**Decision:**

Accept (regular)

**Comment:**

This paper presents ToM-SWE, a dual-agent coding system that augments a software engineering agent with a lightweight user-modeling agent, and evaluates it on Ambiguous SWE-bench, a new Stateful SWE-bench, and a three-week study with professional developers. Reviewers found the problem timely and practically important, and overall the assessment was positive. There are some raised concerns:

* Novelty/framing: the “theory-of-mind” framing is overstated, and that the method is closer to memory or preference modeling.
* Baselines could be stronger, especially against broader memory-based or user-modeling systems.
* Some evaluation choices raised concerns, including GPT-5 as simulator/judge and limited human validation of satisfaction scores.
* Missing ablations and breakdowns, particularly for the memory system and different developer profiles.
* The human study lacks a direct control baseline.

The paper would benefit from a more careful comparison to related memory/user-modeling approaches and slightly more restrained claims around “theory of mind.”